# The interaction between cancer and COVID-19: Risk factors and targeted interventions

**Lucian da Silva Viana**[1]*, **Alython Araujo Chung Filho**[2], **Gina Torres Rego Monteiro**[1],
**Andrea Sobral**[1]

**1** Sergio Arouca National School of Public Health, Oswaldo Cruz Foundation, Rio de Janeiro, Brazil,
**2** Research Center, National Cancer Institute (INCA), Rio de Janeiro, Brazil

* lucianviana@yahoo.com.br

## Abstract

### Background

Cancer patients are particularly fragile, and hospitalization represents a significant additional risk for complications and death, especially given the vulnerability of this group during the global pandemic. The study objective entails analyzing the factors associated with severity and death of cancer patients diagnosed with COVID-19 hospitalized between 2020 and 2022, in the city of Rio de Janeiro, Brazil.

### Materials and Methods

This is a cross-sectional study, with hospitalized patients with both diagnoses, cancer and COVID-19, through two national databases. Bivariate and multivariate regression analyses were performed, followed by an interaction analysis between the variables of interest in the multivariate regression.

### Findings

A total of 1,336 cases were analyzed. Have comorbidity and hematological neoplasms were associated with mechanical ventilation (OR = 1.62 and OR = 2.14, respectively) and with ICU admission (OR = 1.44 and OR = 2.25, respectively). The age group of 60 years or older (OR = 1.33), have multimorbidity (OR = 1.38), and use mechanic ventilation (OR = 2.39) were associated with a higher risk of death from COVID-19. The multimorbidity had a synergistic effect with hospitalization before vaccination (AP = 0.29; 95%CI = 0.01 – 0.58) and the use of mechanical ventilation (AP = 0.54; 95%CI = 0.18 – 0.98).

OPEN ACCESS

**Data availability statement:** There are ethical restrictions which prevent the sharing of minimal data for this study. The data contain potentially identifiable patient information, and must be requested from the Research Ethics Committee (CEP) of the Municipal Health Department of Rio de Janeiro - SMS/RJ through an addendum to the project "Prediction of severity and death due to COVID-19 in cancer patients hospitalized in Rio de Janeiro, between 2020-2022: approach using machine learning" (opinion: 5,837,088). Data are available upon request from the CEP - SMS/RJ through their website (https://saude.prefeitura.rio/comite-de-etica-em-pesquisa), or by contacting them via telephone (+55(21) 2215-1485) or email (cepsmsrj@yahoo.com.br; cepsms@rio.rj.gov.br), for researchers who meet the criteria for access to confidential data.

**Funding:** This work was carried out with support from the Coordination for the Improvement of Higher Education Personnel – Brazil (CAPES) the LSV (001). This study was also funded by the Vice-Directorate of Research and Innovation (VDPI), of the National School of Public Health (ENSP), FIOCRUZ, Rio de Janeiro, in the form of a publication fee. The funders had no role in the study design, data collection, data analysis, data interpretation, or writing of the report. All authors had full access to the study data and were responsible for the decision to submit it for publication.

**Competing interests:** The authors have declared that no competing interests exist.

## Interpretation

These findings are crucial to guide clinical decisions, improve the management of care for these patients, and optimize the use of health resources in the epidemiological context imposed by the pandemic.

## Introduction

In recent years, the global public health scenario has been profoundly impacted by an epidemiological emergency that began in December 2019 in the city of Wuhan, China, and rapidly spread throughout the world [1]. The cause of this crisis was identified as the severe acute respiratory syndrome coronavirus 2 (SARS-CoV-2), designated by the World Health Organization (WHO) in February 2020 [2]. This new disease, known as coronavirus disease 2019 (COVID-19), manifests itself in a variety of ways, including respiratory syndromes, pneumonia, and a range of other complications [3].

Of the more than 1.3 million laboratory-confirmed cases of COVID-19 in the United States between January and May 2020, 14% were hospitalized, 2% were admitted to the intensive care unit (ICU) and 5% died. The incidence was 403.6 cases per 100,000 population, with the highest incidence reported among elderly individuals ≥ 80 years of age (902 cases per 100,000 population) and the lowest among children aged ≤ 9 years (51 cases per 100,000 population). Hospitalizations and deaths were, respectively, six times (45.4% versus 7.6%) and twelve times (19.5% versus 1.6%) higher among patients with at least one reported underlying clinical condition [4], such as cancer. The international [5, 6] and local literature [7, 8] on the subject demonstrates a wide variation in hospital lethality, with different risk factors for severity and death in the oncology population.

Oncology patients are often older, have other coexisting medical conditions, and may have their immune systems compromised due to intensive treatments and the neoplastic disease itself [9–12]. Therefore, hospitalization represents a significant additional risk for serious complications and mortality, especially given the vulnerability of this group during the global pandemic. In addition, there are additional challenges related to the difficulty in confirming the diagnosis and treating cancer, due to the overload of health systems caused by the pandemic [13].

In this context, we hypothesize that the risk for severity and death in patients with cancer and COVID-19 may be modulated by factors such as age, presence of comorbidity, type of tumor and vaccination period, and that such variables may present synergy or antagonism. To support targeted interventions is crucial to investigate hospitalizations of cancer patients due to COVID-19 to understand the risk factors and outcomes of these individuals. This analysis is essential to improve the management and care of these patients, prevent infections, support medical decisions and plan health resources. Thus, this study aims to identify the profile, risk factors for severity and death and their interactions in cancer patients with COVID-19, treated in hospitals in the city of Rio de Janeiro, located in the state of Rio de Janeiro, Brazil, during the first two years of the pandemic.

## Materials and methods

This is a cross-sectional study that identified the clinical and epidemiological profile and analyzed the factors associated with the severity and death of patients diagnosed with cancer and COVID-19, treated in the city of Rio de Janeiro, during the first two years of the pandemic: from March 2020 to February 2022, using two secondary databases, the Hospital Information System (HIS) and the Influenza Epidemiological Surveillance Information System/Severe Acute Respiratory Syndromes (IESIS/SARS).

The city of RJ has an area of 1,200.329 km². In 2022, the population was 6,211,223 inhabitants and the population density was 5,175 inhabitants/km². It has 257 public health facilities in the UHS, of which 15 are registered as specialized oncology care units. This study involved 29 hospital centers, of which 12 are specialized in oncology, where 93% of hospitalizations occurred.

The HIS and IESIS are national systems developed by the Brazilian Ministry of Health. The HIS stores data on hospital admissions within the UHS, reported monthly by all public health facilities, consolidated by municipalities and states, and then sent to the UHS Information Technology Department. IESIS stores all reported cases of hospitalized SARS or any deaths from SARS, within 24 hours, by registered units (hospitals, Emergency Care Units, Emergency Medical Service, Death Verification Service) and forwarded by municipalities and states to the Health Surveillance Secretariat. The consolidated information from both databases can also be made available for research throughout Brazil.

For the sample, a HIS database containing 843,006 patients hospitalized in the city of RJ was made available by the Municipal Health Department (MHD). The procedure for obtaining data involved delimiting the study period (March 1, 2020 to February 28, 2022) (n = 444,138) and excluding patients under 18 years of age (n = 378,222), since cases of severity and death due to COVID-19, hospitalization due to COVID-19 (between 2.5% and 4.1%) and, even rarer, the need for intensive care treatment (below 1%) were uncommon in this age group [14]. After these exclusions, cancer cases were identified using the International Statistical Classification of Diseases and Related Health Problems – 10th Revision (ICD-10), using the code related to the disease in group C, Neoplasms (chapter II), in the variables related to primary diagnoses, or the first secondary oncological diagnosis, when this had not been mentioned in the primary diagnosis (n = 40,597). And patients diagnosed with COVID-19 were identified using the code related to SARS-CoV-2 infection (ICD-10 B34.2) (n = 29,559). After the crossover, a low number of hospitalizations was obtained (n = 129).

In order to identify the real number of patients with both diagnoses hospitalized during the research period and to formulate an integrated information base, the HIS data containing hospitalizations for cancer were linked to the IESIS database, also made available by MHD, containing 208,861 cases. This database was already limited by age and study period. Thus, after excluding lines with empty boxes or those that did not comply with the standardization of the Individual Registration Form for SARS (Data Dictionary: https://www.saude.go.gov.br/files/vigilancia/epidemiologica/fichas-de-notificacao/S/dicionariodedados.pdf) (n = 154,132), the identification of SARS cases due to COVID-19 was carried out using the variable "Final classification" (CLASSI_FIN), which identifies the final diagnosis of the case (code 5 - COVID-19) and the variable "Evolution" (EVOLUCAO), which identifies deaths due to COVID-19 (code 2). Thus, only confirmed cases of COVID-19 were obtained (n = 109,160).

The database integration process was performed using the Python programming language (version 3.10.12) and the Jupyter Notebook development environment (6.4.8), using the following link variables: "Patient name" (HIS: AH_PACIENTE_NOME and IESIS: NM_PACIENT), "Patient's mother's name" (HIS: AH_PACIENTE_NOME_MAE and IESIS: NM_MAE_PAC) and "Patient's date of birth" (HIS: AH_PACIENTE_DT_NASCIMENTO and IESIS: DT_NASC). The Individual Taxpayer Registry, National Health Card and hospital registration number were not fully present in the databases. The latter may change depending on the place of hospitalization, which could generate data duplication. It is worth mentioning that the need for integration between the HIS and IESIS databases arose from the absence of data related to the diagnosis of COVID-19 in the HIS, observed by the discrepancy between this diagnosis in the two databases. The relationship between the databases then made it possible to obtain data that ensured both diagnoses, cancer and COVID-19.

After the integration of the banks, the new database was processed according to the independent variables made available from the Hospital Admission Authorization of the HIS and the Individual SRAG Registration Form of the IESIS. To this end, information on sociodemographic aspects (sex assigned at birth, age group, self-reported race and education level), comorbidities (cardiovascular diseases, respiratory diseases, diabetes mellitus, immunodeficiency or immunosuppression, obesity, etc.), signs and symptoms of COVID-19 (fever, cough, dyspnea, sore throat, etc.), vaccination against COVID-19 (from March 2021 to February 2022), hospital (nosocomial) infection by SARS-CoV-2, type of diagnostic confirmation of COVID-19 (laboratory, imaging, clinical or clinical-epidemiological), ventilatory support (invasive and non-invasive), ICU admission and reason for hospital discharge (discharge, death from COVID-19 and death from other causes) were used. The following dependent variables were used as outcomes for severity: MV (based on the Ventilatory Support variable) and ICU. And for the death outcome, the category Death from COVID-19 was used (based on the Hospital Discharge variable).

## Statistical analysis

To estimate the sample size necessary for association analyses, the method proposed by Wang & Ji [15] was used. For power = 0.8, significance level = 0.05, frequency of the outcome among the not exposed group = 0.5, and OR = 1.2, the required sample number was 816. To calculate the required sample size For interaction analyses, the Excel spreadsheet provided by VanderWeele [16] was used. Was used power = 0.8, significance level = 0.05, frequency of the outcome among the not exposed group = 0.5, OR for each individual effect = 1.5, interaction OR = 1.5, and frequency of each exposure = 0.6. For these parameters, the sample size required is 1204.

A descriptive analysis of cases and deaths was performed, using measures of frequency, central tendency and dispersion. Subsequently, an analysis of the association between clinical and demographic factors with the outcomes of interest (MV, ICU and death from COVID-19) was performed. The exposure variables analyzed were: (1) sex, male or female; (2) race/color, white or non-white; (3) age, <60 years or $\geq$60 years; (4) comorbidity, yes or no; (5) multimorbidity (two or more comorbidities), yes or no; (6) tumor type, hematological or solid; and (7) vaccination period, before or after vaccination. In the analysis for death from COVID-19, also were included as exposure variables, ventilation support, mechanical ventilation and ICU. Initially, a bivariate analysis was conducted, and then, was performed a multivariate logistic regression model including possible confounders. The criteria for inclusion of variables in the adjusted model were: (1) be associated with exposure and outcome in our analysis or in the literature [17–22] (2) Not have collinearity with other included variables (3) not be a consequence of the analyzed exposure. This allowed identifying risk associations in the analyzed outcomes, calculating Odds Ratio (OR) and 95% Confidence Interval (95%CI). A p-value < 0.05 was considered statistically significant.

Finally, an interaction analysis between the factors associated with the outcomes in the multivariate analysis was performed. First, the isolated individual effects of each variable were measured, three measures of interaction in additive scale: Relative Excess Risk due to Interaction (RERI); Attributable Proportion (AP) and Synergy Index (SI). The 95% confidence intervals for RERI, AP and S were estimated using the delta method, using an Excel spreadsheet provided by Andersson et al [21]. The regression analyses were performed using IBM SPSS Statistics® software and the creation of forest plots using GraphPad Prism® software.

## Ethical considerations

The study complies with Resolution Nº. 466/2012 and Resolution Nº. 510/2016, which deal with the rules related to research involving human beings in Brazil, including the waiver of informed consent to participate in cases where obtaining informed consent is not feasible, such as in the use of databases retrospective study of medical records.

The study did not involve experiments on humans and/or the use of human tissue samples, but rather information from two restricted-access databases (Hospital Information System - HIS and Influenza Epidemiological Surveillance

Information System/ Severe Acute Respiratory Syndromes - IESIS/SARS) that were granted by the Rio de Janeiro Municipal Health Department, after submission for review and approval by the Research Ethics Committee (REC) of the Sergio Arouca National School of Public Health, Oswaldo Cruz Foundation (under Opinions Nº. 5592547 and Nº. 6434508) and by the REC of the Rio de Janeiro Municipal Health Department (under Opinions Nº. 5837088 and Nº. 6537856). As this is sensitive informations we are not allowed to provide it.

## Results

There were 1,336 cases of cancer and COVID-19 (Fig 1), of which 40% had two or more hospitalizations. Among the patients hospitalized for cancer, 856 died from COVID-19.

It is noted that the number of cases increased as the age group increased: 9.7% between 18 and 39 years old, 28.7% between 40 and 59 years old and 61.6% in the age group of 60 years or older, with a median age of 62 years (1st/3rd IIQ: 54/72 years) at the time of hospitalization. Among the deaths from COVID-19, this proportion was repeated with 65.0% in the age group of 60 years or older. Females were predominant (57.6% and 56.0%), over males (42.4% and 44.0%), both in cases and deaths, respectively. Non-white races were slightly more frequent (40.3%). Solid tumors were more frequent (88.4%), in relation to hematological neoplasms (11.6%). In 51% of cases, there was one or more comorbidities associated with non-oncologic diseases. Chronic Cardiovascular Diseases - CVD were the most common (33%), followed by Immunodeficiencies or Immunosuppressions (26%) and Diabetes Mellitus – DM (17%). The most common criterion for diagnosing COVID-19 was laboratory (69.7%), followed by clinical epidemiological criteria (19.7%) and image-based criteria (9.8%). The most frequently observed signs and symptoms were: oxygen saturation <95% (49.9%), dyspnea (49.4%), cough (42.2%), respiratory distress (40.9%) and fever (36.2%). It is possible to observe that approximately 47% of cases and deaths required ventilatory support - VS (invasive or not), but only 20% were admitted to the ICU. Hospital deaths

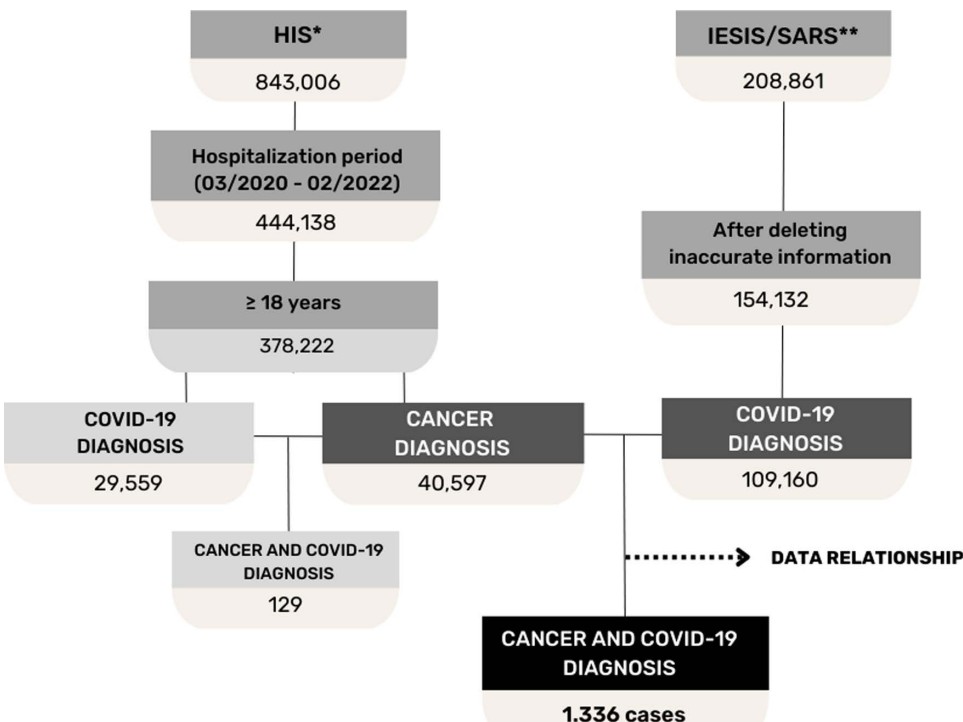

**Fig 1. HIS and IESIS/SRAG banks integration process and case selection.**

due to COVID-19 among cancer patients corresponded to approximately 64% (mortality due to COVID-19). There was an increase to 65% with deaths due to other causes (mortality due to all causes) (Table 1). The mortality rate for solid tumors was higher (64.4%) than for hematological neoplasms (58.4%).

In the multivariate analysis, considering the severity outcomes, have one or more comorbidity and have hematological neoplasms was associated with mechanical ventilation (OR = 1.62, p-value < 0.01; and OR = 2.14, p-value < 0.01, respectively) and with ICU admission (OR = 1.44, p-value = 0.01; and OR = 2.25, p-value < 0.01, respectively) (Table 2).

Regarding the death outcome, the age group of 60 years or older (OR = 1.33; p-value = 0.04), have multimorbidity (OR = 1.38; p-value = 0.02), and use mechanic ventilation (OR = 2.39; p-value < 0.01) were associated with a higher risk of death from COVID-19. On the other hand, have hematological neoplasms (OR = 0.60; p-value = 0.02) and having been hospitalized after vaccination (OR = 0.05; p-value < 0.01) were protection factors (Table 3).

In the interaction analysis, the combination of multimorbidity and having solid tumors had an inhibitory effect (SI = 0.54; 95%CI = 0.29 - 0.99) on death from COVID-19, suggesting that the effect of multimorbidity is stronger in patients with hematological neoplasms. On the other hand, multimorbidity had a synergistic effect with hospitalization before vaccination (AP = 0.29; 95%CI = 0.01 – 0.58) and the use of mechanical ventilation (AP = 0.54; 95%CI = 0.18 – 0.98), indicating that the vaccination protected mainly patients with multimorbidity, and that patients who require mechanical ventilation are at stronger risk if they have multimorbidity (Figure 2). Other interaction between severity and death from COVID-19 can be seen in the Supplementary Table.

## Discussion

The scientific literature [17–21] indicates that cancer patients, due to their often advanced age structure, have other risk factors that, combined with cancer, make them more susceptible to SARS-CoV-2 infection and more severe COVID-19. Other risk factors include respiratory, cardiovascular, renal, and hepatic diseases, obesity, diabetes, and other immuno-suppressive conditions. In this study, the majority of patients were elderly (62%), aged between 60 and 102 years old. Approximately half (51%) of the patients had some other comorbidity that put them at risk for COVID-19, in addition to cancer; 63% of these patients had two or more comorbidities, similar to other studies [18,22], with a high prevalence of CVD (33%) and DM (17%). A meta-analysis found that in patients with COVID-19, the prevalence of hypertension, DM, heart disease, and cerebrovascular disease ranged from 9% to 17% [19]. In the two cohorts, the most common comorbidities were CVD and DM [20–21].

In a study conducted in the first four months of the pandemic by Pontes et al [23] in the state of Paraná, with patients hospitalized for COVID-19, having a comorbidity and this being a CVD increased the risk of death by 8.7 (p = 0.01) and 7.1 (p < 0.01) times, respectively. CVD and cancer are among the most prevalent diseases and are the two leading causes of death worldwide, despite significant improvements in the prevention, diagnosis, and treatment of both diseases [24]. The prior existence of CVD is identified as one of the main risk factors for the incidence and severity of COVID-19. The risk of adverse outcomes is even more pronounced for individuals facing both CVD and cancer diagnoses, especially among the elderly population [25–26].

Since this was a study with cancer patients, immunodeficiency conditions and immunosuppression states were other frequently observed risk factors (26%). There is a close relationship between the immune system and the pathogenesis of cancer, having, among other functions, the role of monitoring, recognizing and eliminating cells with tumorigenic potential. Thereby cancer is formed when cancer cells develop mechanisms to evade the immune system and, at the same time, are capable of promoting a pro-inflammatory state [27]. Inflammation provides growth factors and pro-angiogenic factors that helps tumor proliferation and metastasis [28]. In addition, cancer patients undergo treatments that can cause temporary bone marrow depletion, contributing to immunosuppression [29]. For example, receiving chemotherapy immediately before the onset of COVID-19 symptoms was associated with clinical worsening and death in hospitalized patients [7, 18, 30]. However, covariates related to comorbidities such as immunodeficiency/immunosuppression

**Table 1. Epidemiological and clinical aspects of cases and deaths.**

| | Case | | Death | |
|---|---|---|---|---|
| | n | % | n | % |
| EPIDEMIOLOGICAL ASPECTS | | | | |
| **Age range** | | | | |
| <60 years | 513 | 38.4 | 300 | 35.0 |
| ≥60 years | 823 | 61.6 | 556 | 65.0 |
| **Sex** | | | | |
| Masculine | 567 | 42.4 | 377 | 44.0 |
| Feminine | 769 | 57.6 | 479 | 56.0 |
| **Race/color** | | | | |
| Whites | 523 | 39.1 | 357 | 41.7 |
| Non-white | 539 | 40.3 | 353 | 41.2 |
| No informations | 274 | 20.5 | 146 | 17.1 |
| **Education** | | | | |
| No education | 18 | 1.3 | 15 | 1.8 |
| Basic cycle | 329 | 24.6 | 257 | 30.0 |
| High school | 209 | 15.6 | 165 | 19.3 |
| University education | 48 | 3.6 | 32 | 3.7 |
| No informations | 732 | 54.8 | 387 | 45.2 |
| CLINICAL ASPECTS | | | | |
| **Tumor type** | | | | |
| Solid tumors | 1182 | 88.4 | 766 | 89.5 |
| Hematological neoplasms | 154 | 11.6 | 90 | 10.5 |
| **Comorbidity** | | | | |
| Yes one | 252 | 18.8 | 158 | 18.5 |
| Yes, more than one | 432 | 32.3 | 282 | 32.9 |
| No | 652 | 48.8 | 416 | 48.6 |
| **COVID-19 vaccine*** | | | | |
| Yes | 13 | 2.7 | 5 | 1.9 |
| No | 85 | 17.7 | 39 | 15.3 |
| No informations | 382 | 79.5 | 210 | 82.6 |
| **COVID-19 diagnostic confirmation** | | | | |
| Laboratory | 931 | 69.7 | 532 | 62.1 |
| Others | 395 | 29.6 | 317 | 37.0 |
| No informations | 10 | 0.7 | 7 | 0.8 |
| **Nosocomial infection** | | | | |
| Yes | 104 | 7.8 | 63 | 7.4 |
| No | 613 | 45.9 | 368 | 43.0 |
| No informations | 619 | 46.3 | 425 | 49.6 |
| **Ventilatory support** | | | | |
| Yes, invasive (MV) | 157 | 11.8 | 121 | 14.1 |
| Yes, non-invasive | 471 | 35.3 | 279 | 32.6 |
| No | 388 | 29.0 | 230 | 26.9 |
| No informations | 370 | 27.7 | 226 | 26.4 |
| **ICU admission** | | | | |
| Yes | 266 | 19.9 | 172 | 20.1 |
| No | 797 | 59.7 | 485 | 56.7 |

*(Continued)*

**Table 1.** (Continued)

| | Case | | Death | |
|---|---|---|---|---|
| | n | % | n | % |
| No informations | 273 | 20.4 | 199 | 23.2 |
| **Reason for leaving the hospital** | | | | |
| Discharge due to cure | 356 | 26.6 | – | – |
| Death from COVID-19 | 856 | 64.1 | – | – |
| Death for other reasons | 15 | 1.1 | – | – |
| No informations | 109 | 8.2 | – | – |

*March 2021 to February 2022.

conditions may contain poor record completion, since this is a collection based on self-reporting, which depends on the patient's perception, and may or may not be perceived by them as a factor related to the oncological disease and its treatments.

In the city of São Paulo - SP, Brazil, cancer patients hospitalized for COVID-19 had their deaths associated with the diagnosis of lung cancer (OR: 4.0; p = 0.01), age over 60 years (OR: 4.0; p = 0.01), duration of treatment (OR: 2.7; p = 0.01) or palliative care (OR: 17.6; p < 0.01) for oncological disease during the hospitalization period [7]. In another study, in the city of RJ, between April and May 2020, death was associated with: age over 60 years (60-74 years, OR: 3.6; p < 0.01 and over 75 years, OR: 4.7; p < 0.01), metastatic cancer (p < 0.01), two (OR: 10.4; p < 0.01) or more (OR: 14.2; p < 0.01) sites of metastases, the presence of lung (OR: 8.6; p < 0.01) and bone (OR: 4.3; p < 0.01) metastases, exclusive supportive treatment/palliative care (OR: 4.7; p < 0.01), higher levels of C-reactive protein (OR: 1.0; p < 0.01), hospital admission due to COVID-19 (OR: 2.4; p < 0.01) and use of antibiotics during hospitalization (OR: 3.3; p = 0.02) [8]. In this last study, there was no association between comorbidities and ICU admission with death, similar to the current study and the international literature [6, 9, 30, 31]. In both studies, there was also an association between advanced age and death. However, there was no association between the need for ventilatory support therapies and death, unlike what was observed here.

In our analyses, interestingly, hematological neoplasms were associated with a higher frequency of ICU admission and receipt of MV, however, this group had a lower risk of death. Higher severity risk among patients with hematological malignancies was described by Khawaja et al. [32] in the United States, however, in this study, these patients also had higher death risk. In our cohort, it is possible that these patients had greater access to the ICU and MV when they needed it, and therefore had better results.

This study reveals innovative results on the interaction between risk factors for death in patients with COVID-19 and cancer, especially regarding the role of multimorbidity specific conditions. Although the presence of multimorbidity and solid tumors are both risk factors, they have an inhibitory interaction. This suggests that multimorbidity has a greater effect on the group of patients with hematological malignancies. This may reflect a different profile of comorbidities in both groups, such as the higher frequency of immunosuppression among patients with hematological malignancies. Multimorbidity had a synergistic effect with non-vaccination. This suggests that vaccination has primarily protected patients with multimorbidity. It is possible that the frequency of complete vaccination was higher among patients with multimorbidity, similarly to what was described in hospitalized patients in Italy by Monforte *et al.* [33]. or that vaccination provides greater protection among individuals with more risk factors, similarly to described in the United States by Baker *et al.*, [34] who observed greater protection among obese and older patients. The synergistic effect of multimorbidity with the need for MV can be explained by the higher frequency of hematological malignancies among patients who received MV.

The literature indicates MV in oncology patients admitted to the ICU as an important risk factor for death, and the non-need for MV reduces in-hospital mortality [35]. Another study showed that cancer patients are 3.5 times

**Table 2. Analysis of factors associated with severity outcomes from COVID-19.**

| | Yes (%) | No (%) | Bivariate analysis | | | Multivariate analysis | | |
|---|---|---|---|---|---|---|---|---|
| | | | OR | 95%CI | *P* | OR | 95%CI | *P* |
| MECHANICAL VENTILATION | | | | | | | | |
| **Sex** | | | | | | | | |
| Male | 68 (43.3) | 357 (41.6) | | 1.00 | | | 1.00 | |
| Female | 89 (56.7) | 502 (58.4) | 0.93 | (0.66-1.31) | 0.68 | 1.03 | (0.72 - 1.46) | 0.88[1] |
| **Race/color** | | | | | | | | |
| White | 65 (49.6) | 342 (50.1) | | 1.00 | | | 1.00 | |
| Non-white | 66 (50.4) | 341 (49.9) | 1.01 | (0.70 - 1.48) | 0.92 | 0.96 | (0.65 - 1.40) | 0.83[1] |
| **Age group** | | | | | | | | |
| <60 years | 50 (31.8) | 328 (38.2) | | 1.00 | | | 1.00 | |
| 60 years or older | 107 (68.2) | 531 (61.8) | 1.32 | (0.92 - 1.90) | 0.13 | 1.39 | (0.96 - 2.00) | 0.08[2] |
| **Comorbidity** | | | | | | | | |
| No | 54 (34.4) | 407 (47.4) | | 1.00 | | | 1.00 | |
| Yes | 103 (65.6) | 452 (52.6) | 1.72 | (1.20 - 2.45) | **<0.01** | 1.62 | (1.13 - 2.31) | **<0.01**[3] |
| **Multimorbidity** | | | | | | | | |
| No | 80 (51) | 495 (57.6) | | 1.00 | | | 1.00 | |
| Yes | 77 (49) | 364 (42.4) | 1.31 | (0.93-1.84) | 0.12 | 1.25 | (0.89 - 1.77) | 0.20[3] |
| **Tumor type** | | | | | | | | |
| Solid | 130 (82.8) | 783 (91.2) | | 1.00 | | | 1.00 | |
| Hematogenic | 27 (17.2) | 76 (8.8) | 2.14 | (1.33 - 3.45) | **<0.01** | 2.12 | (1.31 - 3.45) | **<0.01**[4] |
| **Vaccination period** | | | | | | | | |
| Before vaccination | 100 (63.7) | 528 (61.5) | | 1.00 | | | 1.00 | |
| After vaccination | 57 (36.3) | 331 (38.5) | 0.91 | (0.64 - 1.29) | 0.6 | 0.9 | (0.63 - 1.28) | 0.55[5] |
| ICU | | | | | | | | |
| **Sex** | | | | | | | | |
| Male | 113 (42.5) | 332 (41.7) | | 1.00 | | | 1.00 | |
| Female | 153 (57.5) | 465 (58.3) | 0.97 | (0.73 - 1.28) | 0.81 | 1.05 | (0.79 - 1.40) | 0.73[1] |
| **Race/color** | | | | | | | | |
| White | 96 (44.4) | 320 (51) | | 1.00 | | | 1.00 | |
| Non-white | 120 (55.6) | 307 (49) | 1.30 | (0.95 - 1.78) | 0.09 | 1.28 | (0.93 - 1.75) | 0.13[1] |
| **Age group** | | | | | | | | |
| <60 years | 94 (35.3) | 312 (39.1) | | 1.00 | | | 1.00 | |
| 60 years or older | 172 (64.7) | 485 (60.9) | 1.18 | (0.88 - 1.57) | 0.27 | 1.23 | (0.92 - 1.64) | 0.17[2] |
| **Comorbidity** | | | | | | | | |
| No | 101 (38) | 381 (47.8) | | 1.00 | | | 1.00 | |
| Yes | 165 (62) | 416 (52.2) | 1.49 | (1.13 - 1.99) | **<0.01** | 1.44 | (1.08 - 1.92) | **0.01**[3] |
| **Multimorbidity** | | | | | | | | |
| No | 147 (55.3) | 454 (57) | | 1.00 | | | 1.00 | |
| Yes | 119 (44.7) | 343 (43) | 1.07 | (0.81 - 1.42) | 0.63 | 1.05 | (0.79 - 1.39) | 0.74[3] |
| **Tumor type** | | | | | | | | |
| Solid | 220 (82.7) | 730 (91.6) | | 1.00 | | | 1.00 | |
| Hematogenic | 46 (17.3) | 67 (8.4) | 2.28 | (1.52 - 3.41) | **<0.01** | 2.25 | (1.50 - 3.39) | **<0.01**[4] |
| **Vaccination period** | | | | | | | | |
| Before vaccination | 157 (59) | 493 (61.9) | | 1.00 | | | 1.00 | |
| After vaccination | 109 (41) | 304 (38.1) | 1.13 | (0.85 - 1.49) | 0.41 | 1.12 | (0.84 - 1.50) | 0.43[5] |

1 - adjusted for: age group, comorbidity, tumor type and vaccination period.

*(Continued)*

**Table 2.** (Continued)

2 - adjusted for: tumor type and vaccination period.

3 - adjusted for: age group, tumor type, vaccination period.

4 - adjusted for: age group, comorbidity, vaccination period.

5 - adjusted for: age group, comorbidity, tumor type.

**Table 3. Analysis of factors associated with death from COVID-19.**

| | Yes (%) | No (%) | Bivariate analysis | | | Multivariate analysis | | |
|---|---|---|---|---|---|---|---|---|
| | | | OR | 95%CI | *P* | OR | 95%CI | *P* |
| DEATH FROM COVID-19 | | | | | | | | |
| **Sex** | | | | | | | | |
| Male | 377 (44) | 190 (39.6) | | 1.00 | | | 1.00 | |
| Female | 479 (56) | 290 (60.4) | 0.83 | (0.66 - 1.05) | 0.11 | 0.89 | (0.68 - 1.16) | 0.38[1] |
| **Race/color** | | | | | | | | |
| White | 357 (50.3) | 166 (47.2) | | 1.00 | | | 1.00 | |
| Non-white | 353 (49.7) | 186 (52.8) | 0.88 | (0.68 - 1.14) | 0.34 | 0.77 | (0.57 - 1.04) | 0.09[1] |
| **Age group** | | | | | | | | |
| <60 years | 300 (35) | 213 (44.4) | | 1.00 | | | 1.00 | |
| 60 years or older | 556 (65) | 267 (55.6) | 1.48 | (1.18 - 1.86) | **<0.01** | 1.33 | (1.02 - 1.74) | **0.04**[2] |
| **Comorbidity** | | | | | | | | |
| No | 416 (48.6) | 236 (49.2) | | 1.00 | | | 1.00 | |
| Yes | 440 (51.4) | 244 (50.8) | 1.02 | (0.82 - 1.28) | 0.84 | 1.02 | (0.78 - 1.33) | 0.87[3] |
| **Multimorbidity** | | | | | | | | |
| No | 483 (56.4) | 314 (65.4) | | 1.00 | | | 1.00 | |
| Yes | 373 (43.6) | 166 (34.6) | 1.46 | (1.16 - 1.84) | **<0.01** | 1.38 | (1.06 - 1.80) | **0.02**[3] |
| **Tumor type** | | | | | | | | |
| Solid | 766 (89.5) | 416 (86.7) | | 1.00 | | | 1.00 | |
| Hematogenic | 90 (10.5) | 64 (13.3) | 0.76 | (0.54 - 1.08) | 0.12 | 0.60 | (0.39 - 0.91) | **0.02**[4] |
| **Vaccination period** | | | | | | | | |
| Before vaccination | 602 (70.3) | 254 (52.9) | | 1.00 | | | 1.00 | |
| After vaccination | 254 (29.7) | 226 (47.1) | 0.47 | (0.36 - 0.60) | **<0.01** | 0.50 | (0.39 - 0.66) | **<0.01**[5] |
| **Ventilatory support** | | | | | | | | |
| No | 230 (36.5) | 158 (40.9) | | 1.00 | | | 1.00 | |
| Yes | 400 (63.5) | 228 (59.1) | 1.21 | (0.93 - 1.56) | 0.16 | 1.19 | (0.91 -1.55) | 0.21[6] |
| **Mechanical ventilation** | | | | | | | | |
| No | 509 (80.8) | 350 (90.7) | | 1.00 | | | 1.00 | |
| Yes | 121 (19.2) | 36 (9.3) | 2.31 | (1.56 - 3.44) | **<0.01** | 2.39 | (1.59 - 3.59) | **<0.01**[5] |
| **ICU** | | | | | | | | |
| No | 485 (73.8) | 312 (76.8) | | 1.00 | | | 1.00 | |
| Yes | 172 (26.2) | 94 (23.2) | 1.18 | (0.88 - 1.57) | 0.27 | 1.24 | (0.91 - 1.67) | 0.17[6] |

1 – adjusted by: age group, multimorbidity, tumor type, vaccination period, mechanical ventilation.

2 – adjusted by: tumor type, vaccination period, mechanical ventilation.

3 – adjusted by age group, tumor type, vaccination period, mechanical ventilation.

4 - adjusted by age group, multimorbidity, vaccination period, mechanical ventilation.

5 - adjusted by age group, multimorbidity, tumor type, mechanical ventilation.

6 - adjusted by age group, multimorbidity, tumor type, vaccination period.

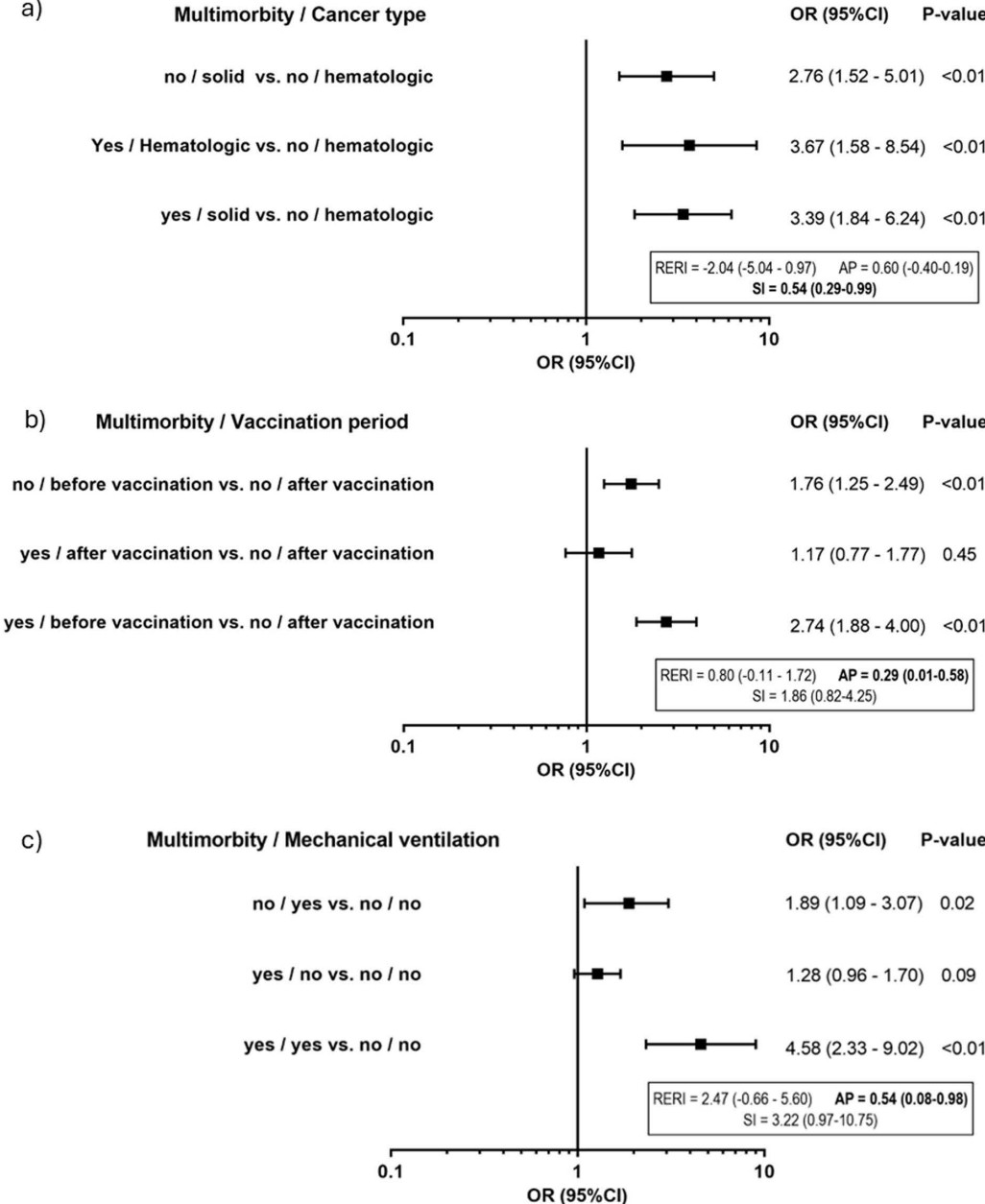

**Fig 2. Interaction of factors associated with death from COVID-19.** a) Multimorbidity and tumor type. b) Multimorbidity and vaccination period. c) Multimorbidity and mechanical ventilation. 1 - Adjusted for: age group, vaccination period and mechanical ventilation. 2 - Adjusted for: age group, tumor type and mechanical ventilation. 3 - Adjusted for: age group, tumor type and vaccination period.

more likely to require MV, be admitted to the ICU, and die than non-cancer patients [36]. One study showed that, in a comparison of COVID-19 patients with and without a cancer diagnosis, cancer patients had a higher mortality rate (OR: 2.34; p = 0.03), a higher number of ICU admissions (OR: 2.84; p < 0.01), and a greater presence of severe symptoms (OR: 2.79; p < 0.01) [17]. Comparisons between patients affected by both diseases and those

hospitalized only with cancer or COVID-19 could reveal additional information, but this goes beyond the scope of this research.

As here, some studies demonstrate a higher frequency of COVID-19 in women [7, 8, 18], but without association with aggressiveness and death. However, there are reports of an association between male gender and death during hospitalization [11, 19]. Information on educational level obtained a low percentage in this study (45.2%), with the majority of patients having completed only basic level of education (24.6%). White and non-white races were similarly frequent (39.1% and 40.3%, respectively). The non-white races are black, brown, Asian and indigenous people, with no representation of the latter two. All these races/colors are defined by the Brazilian Institute of Geography and Statistics (https://www.ibge.gov.br/en/home-eng.html?lang=en-GB). In this study, there was no association between race/color and the outcomes MV, ICU and death from COVID-19. However, new studies may contribute to a more in-depth analysis of the relationship between variables such as race/color and education, with socioeconomic conditions, this factor of great diversity in the Brazilian population may have contributed to morbidity and mortality due to COVID-19 [37], but this is not the objective of this study.

The frequency of COVID-19 signs and symptoms in this study was similar between cases with or without fatal outcome, with the most common symptoms (fever, cough, dyspnea, respiratory distress, and oxygen saturation <95%) also found in other studies that characterize this population [8, 17, 18]. Only 70% of COVID-19 diagnoses were made through laboratory tests. During the pandemic, some countries, including Brazil, adopted restrictive measures regarding the use of laboratory tests, due to budgetary constraints, population size or the government's current plan. This influenced the estimates of the number of people infected and deaths from the SARS-CoV-2 virus [38]. Furthermore, there is also a high rate of nosocomial transmission of SARS-CoV-2 in cancer patients, since they have a greater demand for care in health units for treatment [39]. In this study, nosocomial transmission was confirmed in only 8% of hospitalizations. This percentage may be underestimated, due to the incubation time and asymptomatic clinical manifestation, given the high proportion of hospitalizations without this information (46%), which may generate underreporting of this type of hospital infection. It is worth mentioning that hospital-acquired COVID-19 was associated with increased mortality in another study [31]. Another variable with low completion was vaccination against COVID-19 (20.4%). This may also be underestimated and not represent the reality of the population of cancer patients, since they were encouraged since the beginning of vaccination in RJ, in February 2021, to comply with the vaccination schedule against COVID-19 [40].

Some additional limitations should be mentioned: 1) some variables such as race/color, comorbidities and signs/symptoms of COVID-19 may be under/overestimated, since they are self-reported, that is, based on the total perception of patients. 2) the absence of specific clinical information on the cancer disease, such as tumor grade, disease staging, performance status (degree of functionality), treatments and hospitalizations prior to the diagnosis of COVID-19, which could add additional information, providing more robust results. The use of electronic medical records could aggregate more data, but this is not a reality in all public services and there is no unified digital system in the city of RJ that integrates this information. 3) the lack of a univocal and common identification key for linking databases resulted in operational difficulties and required the use of probabilistic techniques to identify pairs based on character sequences [41]. 5) 40% of patients were hospitalized more than once and it is known that all had at least one COVID-related hospitalization, although the available information does not allow us to state in which hospitalization the SARS-CoV-2 infection occurred, with the diagnostic date being present only in the IESIS database, which prevents more effective integration of data between databases. 6) the poor completion of some variables, such as dates of death and vaccination against the SARS-CoV-2 virus, which could bring greater accuracy to the findings. Furthermore, errors in records, such as in the diagnosis of SARS, or in the coding of cancer diagnoses may have led to false matching of cases, which may also explain the low number of cancer and COVID-19 cases (n = 129) in the HIS database. However, any information that provides a better understanding of diseases and conditions such as COVID-19 is essential for defining priorities and actions aimed at cancer control, as well as for indirectly evaluating the effectiveness of the interventions instituted.

This study is relevant in the context of the COVID-19 pandemic in Brazil, one of the most affected countries globally. It contributes significantly by offering a comprehensive and multicenter analysis (29 health centers), with a large sample size and considering a critical period of the first two years of the pandemic. Another strength of this study is that these findings highlight how the interaction between COVID-19 and cancer can result in serious complications and higher mortality in hospitalized patients, providing crucial information for the management and monitoring of these risk groups during the pandemic, contributing to healthcare planning, resource allocation, and improving the quality of care in possible future similar situations.

In our study, patients with comorbidities and hematological neoplasms had increased risk for UCI and MV. Elderly patients, with solid tumors, with multimorbidity, hospitalized before the start of vaccination, and who required MV, had higher death risk. The presence of multimorbidity preferentially affected patients with hematological neoplasms, hospitalized before vaccination and who required MV. These findings are crucial to guide clinical decisions, improve the management of care for these patients, and optimize the use of health resources in the epidemiological context imposed by the pandemic.

Further studies on the behavior of diseases such as cancer in the context of the COVID-19 pandemic could support improvements in health care, from the targeting of preventive programs to the provision of specialized treatments for this population, in addition to contributing to the international and local-regional debate on health assessment and monitoring and COVID-19 surveillance.

## Supporting information

**S1 Table. Interaction of factors associated with mechanical ventilation, ICU and death from COVID-19.**
(DOCX)

## Author contributions

**Conceptualization:** Lucian da Silva Viana, Andrea Sobral.

**Formal analysis:** Lucian da Silva Viana, Alython Araujo Chung Filho, Gina Torres Rego Monteiro, Andrea Sobral.

**Funding acquisition:** Andrea Sobral.

**Investigation:** Lucian da Silva Viana, Alython Araujo Chung Filho.

**Methodology:** Lucian da Silva Viana, Andrea Sobral.

**Project administration:** Andrea Sobral.

**Supervision:** Andrea Sobral.

**Validation:** Alython Araujo Chung Filho, Andrea Sobral.

**Visualization:** Gina Torres Rego Monteiro.

**Writing – original draft:** Lucian da Silva Viana.

**Writing – review & editing:** Lucian da Silva Viana, Alython Araujo Chung Filho, Gina Torres Rego Monteiro, Andrea Sobral.

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
