## [Decision Letter · Decision Letter 0]

2 Dec 2024

PONE-D-24-30523Factors associated with severity and death in patients with cancer and COVID-19PLOS ONE

Dear Dr. Viana,

Thank you for submitting your manuscript to PLOS ONE. After careful consideration, we feel that it has merit but does not fully meet PLOS ONE’s publication criteria as it currently stands. Therefore, we invite you to submit a revised version of the manuscript that addresses the points raised during the review process.

 This study is cross-sectional as it assesses the prevalence of certain factors and outcomes in cancer patients hospitalized with COVID-19 over a defined period. This information provides a comprehensive view of the clinical and epidemiological profile of cancer patients with COVID-19, highlighting the impact of age, comorbidities, and cancer type on severity and mortality outcomes.

However, the results (advanced age, comorbidities significantly increase the severity and mortality of COVID-19 in cancer patients, ventilatory support and MV are strong indicators of poor outcomes) are expected as are already known in the literature. A strength of the paper might be that offers crucial insights for managing cancer patients during the pandemic and highlights the need for further studies to improve care and health policies for this vulnerable population.  This study's findings are important for healthcare planning, resource allocation, and improving the quality of care during pandemics. A dedicated paragraph in discussion can include the above. A possible more appropriate title could be “the interaction between cancer and COVID-19 including the role of socioeconomic factors and providing targeted interventions.”

Also, important limitations that should mentioned are potential underreporting and overreporting due to self-reported data on race, comorbidities, and symptoms. Lack of detailed clinical information on cancer disease (tumor grade, staging, treatments).  Difficulties in database integration and linking due to data collection methods and incomplete records.

We look forward to receiving your revised manuscript.

Kind regards,

Eleni Magira

Academic Editor

PLOS ONE

Journal Requirements:

1. When submitting your revision, we need you to address these additional requirements. Please ensure that your manuscript meets PLOS ONE's style requirements, including those for file naming. The PLOS ONE style templates can be found at https://journals.plos.org/plosone/s/file?id=wjVg/PLOSOne_formatting_sample_main_body.pdf and https://journals.plos.org/plosone/s/file?id=ba62/PLOSOne_formatting_sample_title_authors_affiliations.pdf 2. Please include captions for your Supporting Information files at the end of your manuscript, and update any in-text citations to match accordingly. Please see our Supporting Information guidelines for more information: http://journals.plos.org/plosone/s/supporting-information.

Additional Editor Comments:

This study is cross-sectional as it assesses the prevalence of certain factors and outcomes in cancer patients hospitalized with COVID-19 over a defined period. This information provides a comprehensive view of the clinical and epidemiological profile of cancer patients with COVID-19, highlighting the impact of age, comorbidities, and cancer type on severity and mortality outcomes.

However, the results (advanced age, comorbidities significantly increase the severity and mortality of COVID-19 in cancer patients, ventilatory support and MV are strong indicators of poor outcomes) are expected as are already known in the literature. A strength of the paper might be that offers crucial insights for managing cancer patients during the pandemic and highlights the need for further studies to improve care and health policies for this vulnerable population. This study's findings are important for healthcare planning, resource allocation, and improving the quality of care during pandemics. A dedicated paragraph in discussion can include the above. A possible more appropriate title could be “the interaction between cancer and COVID-19 including the role of socioeconomic factors and providing targeted interventions.”

Also, important limitations that should mentioned are potential underreporting and overreporting due to self-reported data on race, comorbidities, and symptoms. Lack of detailed clinical information on cancer disease (tumor grade, staging, treatments). Difficulties in database integration and linking due to data collection methods and incomplete records.

Reviewers' comments:

Reviewer's Responses to Questions

**Comments to the Author**

1. Is the manuscript technically sound, and do the data support the conclusions?

Reviewer #1: Yes

2. Has the statistical analysis been performed appropriately and rigorously? 

Reviewer #1: Yes

3. Have the authors made all data underlying the findings in their manuscript fully available?

Reviewer #1: No

4. Is the manuscript presented in an intelligible fashion and written in standard English?

Reviewer #1: Yes

5. Review Comments to the Author

Reviewer #1: 1. Research hypothesis not mentioned.

2. The validation of the questionnaire not done.

3. Conclusion should be further modified to bring in line with the objectives

4. The sample size needs to be elaborated

6. PLOS authors have the option to publish the peer review history of their article (what does this mean? ). If published, this will include your full peer review and any attached files.

**Do you want your identity to be public for this peer review?** For information about this choice, including consent withdrawal, please see our Privacy Policy .

Reviewer #1: **Yes: ** MILI SENGAR

---

## [Author Response · Author response to Decision Letter 0]

6 Feb 2025

1) The results (advanced age, comorbidities significantly increase the severity and mortality of COVID-19 in cancer patients, ventilatory support and MV are strong indicators of poor outcomes) are expected as are already known in the literature. A strength of the paper might be that offers crucial insights for managing cancer patients during the pandemic and highlights the need for further studies to improve care and health policies for this vulnerable population. This study's findings are important for healthcare planning, resource allocation, and improving the quality of care during pandemics. A dedicated paragraph in discussion can include the above.

Reply: The above observations reinforce the contributions of the study, which provides crucial insights for the management of cancer patients during the pandemic. The results obtained are essential for the development of health strategies, the effective allocation of resources and the continuous improvement of the quality of care in possible future pandemics.

To bring even more robustness to the results obtained, three more variables were included in the study (Multimorbidity, Vaccination Period and Tumor Type). There were no changes in Table 1 and Figure 1. Table 2, which concerns severity and death from COVID-19, has become larger, so we divided it into two: Table 2 (severity) and Table 3 (death). Figure 2 has been modified with the new analyses of interactions recently discovered. Other interactions between severity and death from COVID-19 can be seen in the Supplementary Table (Table S1), also added. Therefore, the description of the results and part of the discussion have been modified (including some new references). All tese changes are marked in blue in the text.

2) A possible more appropriate title could be “the interaction between cancer and COVID-19 including the role of socioeconomic factors and providing targeted interventions.”

Reply: We understand that the previous title (“Factors associated with severity and death in patients with cancer and COVID-19”) did not accurately express the main focus of the study. In view of this, the title was revised to “The interaction between cancer and COVID-19: risk factors and targeted interventions”, in order to more clearly reflect the objectives of the study.

3) Important limitations that should mentioned are potential underreporting and overreporting due to self-reported data on race, comorbidities, and symptoms. Lack of detailed clinical information on cancer disease (tumor grade, staging, treatments). Difficulties in database integration and linking due to data collection methods and incomplete records.

Reply: The limitations were carefully highlighted in an extensive paragraph in the “Discussion” topic (paragraph 10, page 16), where the points mentioned by the reviewer were included:

“Some additional limitations should be mentioned: 1) some variables such as race/color, comorbidities and signs/symptoms of COVID-19 may be under/overestimated, since they are self-reported, that is, based on the total perception of patients. 2) the absence of specific clinical information on the cancer disease, such as tumor grade, disease staging, performance status (degree of functionality), treatments and hospitalizations prior to the diagnosis of COVID-19, which could add additional information, providing more robust results. The use of electronic medical records could aggregate more data, but this is not a reality in all public services and there is no unified digital system in the city of RJ that integrates this information. 3) the lack of a univocal and common identification key for linking databases resulted in operational difficulties and required the use of probabilistic techniques to identify pairs based on character sequences [36]. 5) 40% of patients were hospitalized more than once and it is known that all had at least one COVID-related hospitalization, although the available information does not allow us to state in which hospitalization the SARS-CoV-2 infection occurred, with the diagnostic date being present only in the IESIS database, which prevents more effective integration of data between databases. 6) the poor completion of some variables, such as dates of death and vaccination against the SARS-CoV-2 virus, which could bring greater accuracy to the findings. Furthermore, errors in records, such as in the diagnosis of SARS, or in the coding of cancer diagnoses may have led to false matching of cases, which may also explain the low number of cancer and COVID-19 cases (n = 129) in the HIS database. However, any information that provides a better understanding of diseases and conditions such as COVID-19 is essential for defining priorities and actions aimed at cancer control, as well as for indirectly evaluating the effectiveness of the interventions instituted.”

4) Ensure that your manuscript meets PLOS ONE's style requirements, including those for file naming.

Reply: Changes were made in accordance with PLOS ONE style requirements.

5) Include captions for your Supporting Information files at the end of your manuscript, and update any in-text citations to match accordingly.

Reply: Supporting information files were included at the end of the main text (Table S1).

6) Have the authors made all data underlying the findings in their manuscript fully available? If there are restrictions on publicly sharing data—e.g. participant privacy or use of data from a third party—those must be specified.

Reply: “The study complies with Resolution Nº. 466/2012 and Resolution Nº. 510/2016, which deal with the rules related to research involving human beings in Brazil, including the waiver of informed consent to participate in cases where obtaining informed consent is not feasible, such as in the use of databases retrospective study of medical records.

The study did not involve experiments on humans and/or the use of human tissue samples, but rather information from two restricted-access databases (Hospital Information System - HIS and Influenza Epidemiological Surveillance Information System/ Severe Acute Respiratory Syndromes - IESIS/SARS) that were granted by the Rio de Janeiro Municipal Health Department, after submission for review and approval by the Research Ethics Committee (REC) of the Sergio Arouca National School of Public Health, Oswaldo Cruz Foundation (under Opinions Nº. 5592547 and Nº. 6434508) and by the REC of the Rio de Janeiro Municipal Health Department (under Opinions Nº. 5837088 and Nº. 6537856). As this is sensitive informations we are not allowed to provide it.”

This information has been added to the main text in the “Ethical Considerations” subtopic of Materials and Methods (paragraph 11 and 12, page 8).

7) Research hypothesis not mentioned.

Reply: We add the research hypothesis in the “Introduction” topic (paragraph 4, page 3): “In this context, we hypothesize that the risk for severity and death in patients with cancer and COVID-19 may be modulated by factors such as age, presence of comorbidity, type of tumor and vaccination period, and that such variables may present synergy or antagonism.”

8) The validation of the questionnaire not done.

Reply: No questionnaire was applied; patient data were acquired after integration between the following databases: Hospital Information System - HIS and Influenza Epidemiological Surveillance Information System/ Severe Acute Respiratory Syndromes - IESIS/SARS. The database integration process was performed using the Python programming language (version 3.10.12) and the Jupyter Notebook development environment (6.4.8). The complete data acquisition process is described in the Materials and Methods (paragraph 6, page 6).

9) Conclusion should be further modified to bring in line with the objectives.

Reply: We have made the following changes to the text, so that it is in line with our initial objectives (paragraph 11 and 12, page 17).:

“This study is relevant in the context of the COVID-19 pandemic in Brazil, one of the most affected countries globally. It contributes significantly by offering a comprehensive and multicenter analysis (29 health centers), with a large sample size and considering a critical period of the first two years of the pandemic. Another strength of this study is that these findings highlight how the interaction between COVID-19 and cancer can result in serious complications and higher mortality in hospitalized patients, providing crucial information for the management and monitoring of these risk groups during the pandemic, contributing to healthcare planning, resource allocation, and improving the quality of care in possible future similar situations.

In our study, patients with comorbidities and hematological neoplasms had increased risk for UCI and MV. Elderly patients, with solid tumors, with multimorbidity, hospitalized before the start of vaccination, and who required MV, had higher death risk. The presence of multimorbidity preferentially affected patients with hematological neoplasms, hospitalized before vaccination and who required MV. These findings are crucial to guide clinical decisions, improve the management of care for these patients, and optimize the use of health resources in the epidemiological context imposed by the pandemic.”

10) The sample size needs to be elaborated.

Reply:The sample size was calculated and provided in the “Statistical analysis” subtopic of the Materials and Methods (paragraph 8, page 7).

“To estimate the sample size necessary for association analyses, the method proposed by Wang & Ji [15] was used. For power = 0.8, significance level = 0.05, frequency of the outcome among the not exposed group = 0.5, and OR = 1.2, the required sample number was 816. To calculate the required sample size For interaction analyses, the Excel spreadsheet provided by VanderWeele [16] was used. Was used power = 0.8, significance level = 0.05, frequency of the outcome among the not exposed group = 0.5, OR for each individual effect = 1.5, interaction OR = 1.5, and frequency of each exposure = 0.6. For these parameters, the sample size required is 1204.”

Additional questions

1. We notice that your manuscript file was uploaded on July 29 2024. Please can you upload the latest version of your revised manuscript as the main article file, ensuring that does not contain any tracked changes or highlighting. This will be used in the production process if your manuscript is accepted. Please follow this link for more information: http://blogs.PLOS.org/everyone/2011/05/10/how-to-submit-your-revised-manuscript/

Please amend the title either on the online submission form or in your manuscript so that they are identical.

Please amend your authorship list in your manuscript file to include author Alython Araujo Chung Filho.

Answer: Change of the “Manuscript” file, to the most current version of the manuscript, on February 4, 2025, in this version containing the name of the author Alython Araujo Chung Filho. Changed the title on the online submission form.

Answer: There are ethical restrictions which prevent the sharing of minimal data for this study. The data contain potentially identifiable patient information, and must be requested from the Research Ethics Committee (CEP) of the Municipal Health Department of Rio de Janeiro - SMS/RJ through an addendum to the project “Prediction of severity and death due to COVID-19 in cancer patients hospitalized in Rio de Janeiro, between 2020-2022: approach using machine learning” (opinion: 5,837,088). Data are available upon request from the CEP - SMS/RJ through their website (https://saude.prefeitura.rio/comite-de-etica-em-pesquisa), or by contacting them via telephone (+55(21) 2215-1485) or email (cepsmsrj@yahoo.com.br; cepsms@rio.rj.gov.br), for researchers who meet the criteria for access to confidential data.

---

## [Editor Report · Decision Letter 1]

12 Feb 2025

The interaction between cancer and COVID-19: risk factors and targeted interventions

PONE-D-24-30523R1

Dear Dr. Lucian da Silva Viana,

We’re pleased to inform you that your manuscript has been judged scientifically suitable for publication and will be formally accepted for publication once it meets all outstanding technical requirements.

Kind regards,

Eleni Magira

Academic Editor

PLOS ONE

---

## [Editor Report · Acceptance letter]

PONE-D-24-30523R1

PLOS ONE

Dear Dr. Viana,

I'm pleased to inform you that your manuscript has been deemed suitable for publication in PLOS ONE. Congratulations! Your manuscript is now being handed over to our production team.

Kind regards,

on behalf of

Dr. Eleni Magira

Academic Editor

PLOS ONE